# Humans are more prosocial in poor foraging environments

Todd A. Vogel [1,2,3] ✉, Luke Priestley[4], Jo Cutler [1,2,3], Tabitha Hogg[1,2,3], Nima Khalighinejad [4], Neil Garrett[5], Matthew A. J. Apps [1,2,3], Matthew F. S. Rushworth [4] & Patricia L. Lockwood [1,2,3,4] ✉

Prosocial behaviours are essential for solving global challenges. Often, these behaviours have been measured using economic games or tasks where people decide between helping or not. However, in everyday life current behaviours are interrupted with alternative opportunities. Across three independent samples (two preregistered, total $n = 510$), people watched a movie whilst encountering opportunities that benefitted another person or themselves. Crucially, participants decided in different poor and rich foraging environments where the average reward values of opportunities changed. We demonstrate a stronger environmental influence on decisions that benefit others: people were more willing to interrupt their behaviour to help others in poor environments, where the average reward value was lower, compared to richer environments where average reward value was higher. Computational modelling revealed that the opportunity costs of the different foraging environments were valued distinctly for others. Factors of utilitarianism, and empathy/emotional motivation, captured variability in opportunity costs for others. We show that when humans decide to engage in prosocial behaviours depends on the quality of opportunities in one's environment, which is critical as environments change.

Humans constantly decide whether to interrupt what they are doing to try something else. Current experimental paradigms, however, often present people with two options to decide between, which do not reflect these kinds of decisions that people face in everyday life[1–3]. Indeed, in the real world, we regularly come across opportunities to interrupt what we are doing to reward others—we decide to stop watching TV or scrolling through social media to help children with homework or make tea for a partner. Theories from behavioural ecology describing non-human animal behaviour suggest that these types of decisions about when to act depend on the opportunity costs and context of one's environment[4–6]. We evaluate what we are doing now against other opportunities as we encounter them, and are driven to switch when better options appear. This ecological framework is beginning to be used to study human behaviour[1,7–13], yet whether these environmental influences extend to decisions that affect other people is still unknown. This is despite such prosocial behaviours being key to solving global challenges[14] and for societal cohesion broadly[15–17]. Indeed, the nature of altruism is a fundamental question of being human[18]. Does when we decide to help others depend on the richness of our environment?

Human prosocial behaviour is often measured using economic games such as the dictator game, trust game, or the prisoner's dilemma[19–22]. Other studies have compared two-alternative forced choices between exerting effort for rewards or resting[23,24]. While these

[1]Centre for Human Brain Health, School of Psychology, University of Birmingham, Birmingham, UK. [2]Institute for Mental Health, School of Psychology, University of Birmingham, Birmingham, UK. [3]Centre for Developmental Science, School of Psychology, University of Birmingham, Birmingham, UK. [4]Oxford Centre for Integrative Neuroimaging, Department of Experimental Psychology, University of Oxford, Oxford, UK. [5]School of Psychology, University of East Anglia, Norwich Research Park, Norwich, UK. ✉e-mail: t.vogel@bham.ac.uk; p.l.lockwood@bham.ac.uk

paradigms present people with many decisions, they may not capture the types of everyday behaviours and prosocial decisions that our brains evolved to solve[2,13,25,26]. In real-world settings, one's current behaviour is interrupted by a choice to help others, or to do something else more beneficial for oneself. What remains unclear, however, is how different environments affect decisions to act to help other people. This is despite many decisions we make having direct impacts on other people, and not only ourselves[27].

More broadly, research in social psychology has examined the environmental factors that determine when we help others. For example, the presence of others[28–32], the urgency of the situation[33,34], or the perceived costs of acting[35–37] can influence how likely we are to act to help[16]. The impact of the broader environmental context is less clear. The richness of one's social environment—i.e. relationships, social network, etc.—relates to levels of prosocial behaviour[38–42], but the specific impact of one's economic environment is highly debated[43]. Some studies report that those with lower income or financial well-being may be more prosocial[44–46], while others report that those with higher income or financial well-being are more generous[17,47–49]. However, these studies are often correlational and do not experimentally manipulate the environment. Whether poor economic environments lead to higher levels of prosociality, and the mechanisms that ultimately drive decisions about when to help others, are still unknown.

A separate line of evidence from behavioural ecology suggests that in both humans and non-human animals, one's environment strongly influences decisions about when and how to act[3,5,7,12,25]. Theories of 'prey-selection' in animal foraging suggest that the quality, or richness, of an animal's environment determines how it chooses its prey[50,51]. In support, a seminal study in birds showed that in rich environments, where the average quality of prey is higher, animals will reject low-quality prey they encounter and instead selectively wait for higher quality options[51]. Here, the opportunity costs of selecting prey are higher, as alternative options are likely to be of high quality too. When the foraging environment is poor—i.e. where the average quality of prey is lower—waiting for high-quality options can be disadvantageous. The animals will begin to accept low-quality prey because future opportunities are not expected to be better (i.e. the opportunity cost of selecting a prey is less, because better alternatives are not readily available). This environmental influence on deciding when to act has been observed in a wide range of animals, from worms to monkeys[4,6,25] and more recently in humans[9,10]. To date, however, most human studies have focused on decisions within other foraging contexts, such as whether to explore a patch in which one resides, or leave to exploit an alternative patch[52]. Here, we tested whether the richness of one's environment affects when humans interrupt their behaviour to help others and themselves, and whether these decisions conform to predictions from prey-selection theory. We were also interested in whether the environmental influence on decisions to act was stronger or weaker when acting prosocially compared to benefiting oneself. This allowed us to demonstrate whether an overlooked aspect, the quality of opportunities we encounter, could fundamentally shape the willingness to engage in human prosocial behaviour.

We combined well-established theories and computational principles about animal behaviour from behavioural ecology and decision neuroscience[1] to test the impact of poor and rich foraging environments on decisions about when to help others. We defined poor and rich environments based on the relative frequency of low- vs high-quality opportunities that participants encountered. Across 3 separate studies (study 1 $n = 237$, study 2 $n = 219$, study 3 $n = 54$), we adapted a prey selection paradigm where people watched a movie and encountered opportunities to earn potential rewards by exerting a fixed amount of physical effort, calibrated to their own ability, to mimic the kind of energy costs necessary for acting to help another or oneself in everyday life (Fig. 1a). Crucially, on some trials, participants had the chance to interrupt their ongoing behaviour to win rewards for an anonymous other person (prosocial condition). The opportunities signalled the magnitude and probability of a reward, and their frequency differed in the two environments. If the participant decided to act and pursue the opportunity, the movie would disappear while they completed the effort task. If the participant declined the opportunity, they did not have to take any action, and the movie continued playing. Framing the question in this way makes the context-dependent effects analyzable using ecological models[13,53] and better captures how we encounter real-world decisions about when to offer help. We instructed participants that their decisions would be completely anonymous, and that the other player would be unaware that the participant would be earning rewards on their behalf. This allowed the 'other' trials to be identical in all aspects to 'self' trials except for the reward recipient, and controlled for motives such as reciprocity and reputation affecting prosocial decisions.

Across all 3 studies, we show a reliable environmental influence on decisions to act that is stronger when deciding to help others compared to oneself. We found that participants were more likely to interrupt their behaviour in poor foraging environments than in rich ones as the quality of the reward increased, and that this environmental effect was larger when deciding for others. Computational modelling revealed that opportunity costs for self and others were distinctly encoded in the different environments, and that people were equally sensitive to value when deciding for others in a poor environment as for themselves in a rich environment. Distinct factors of utilitarianism and empathy/motivation, but not psychiatric traits, were related to variability in opportunity costs for others. Together, we demonstrate that choosing to act to help others closely depends on the foraging qualities of one's environment, and that this environmental influence is stronger than when deciding to help oneself.

## Results

### People act to help others more in poor compared to rich foraging environments

Participants in study 1 ($n = 237$ participants, aged 18–35 years $M$ ($SD$) = 28.8 (4.4); self-reported gender: 124 women, 112 men, 1 non-binary) completed the Prosocial Ecology Task (Fig. 1). We tested our central preregistered hypothesis (AsPredicted #102887) that the richness of the environment (in terms of the average quality of reward opportunities) would influence decisions to act for others differently from decisions that benefit oneself. To do so, we built a generalised linear mixed-effects model (see "Methods" section) that included environment (poor vs rich), recipient (self vs other), and the expected value of the reward (magnitude*probability) as predictors of choices to act—i.e. to skip part of the movie and pursue potential rewards. We found that, as expected value increased, participants were more likely to choose to act in the poor environment compared to the rich one, and that this difference was greater when the choice benefitted the other person relative to oneself (three-way interaction: odds ratio (OR) = 1.49, 95% confidence interval [1.05, 2.12], $z = 2.20$, $p = 0.028$, Fig. 2a, see Table S1 for full model results). In other words, people decided to act more often in poor foraging environments compared to rich ones, and this environmental influence was stronger when making prosocial decisions than self-benefiting ones.

In line with our preregistration, we ran several control analyses to account for potential effects of fatigue and previously seen opportunities. We first included trial number, as a proxy for fatigue, to the model before also adding the previous trial's choice and expected value. After controlling for these variables, we still found that people were more likely to help others in poor environments (three-way interaction: OR = 1.46 [1.13, 1.88], $z = 2.88$, $p = 0.004$). We also observed that participants were less likely to choose to act over time (OR = 0.52 [0.48, 0.57], $z = 14.95$, $p < 0.001$), but were more likely to act if they had done so on the previous opportunity (OR = 1.22 [1.10, 1.36], $z = 3.63$, $p < 0.001$). There was no statistically significant effect of the expected

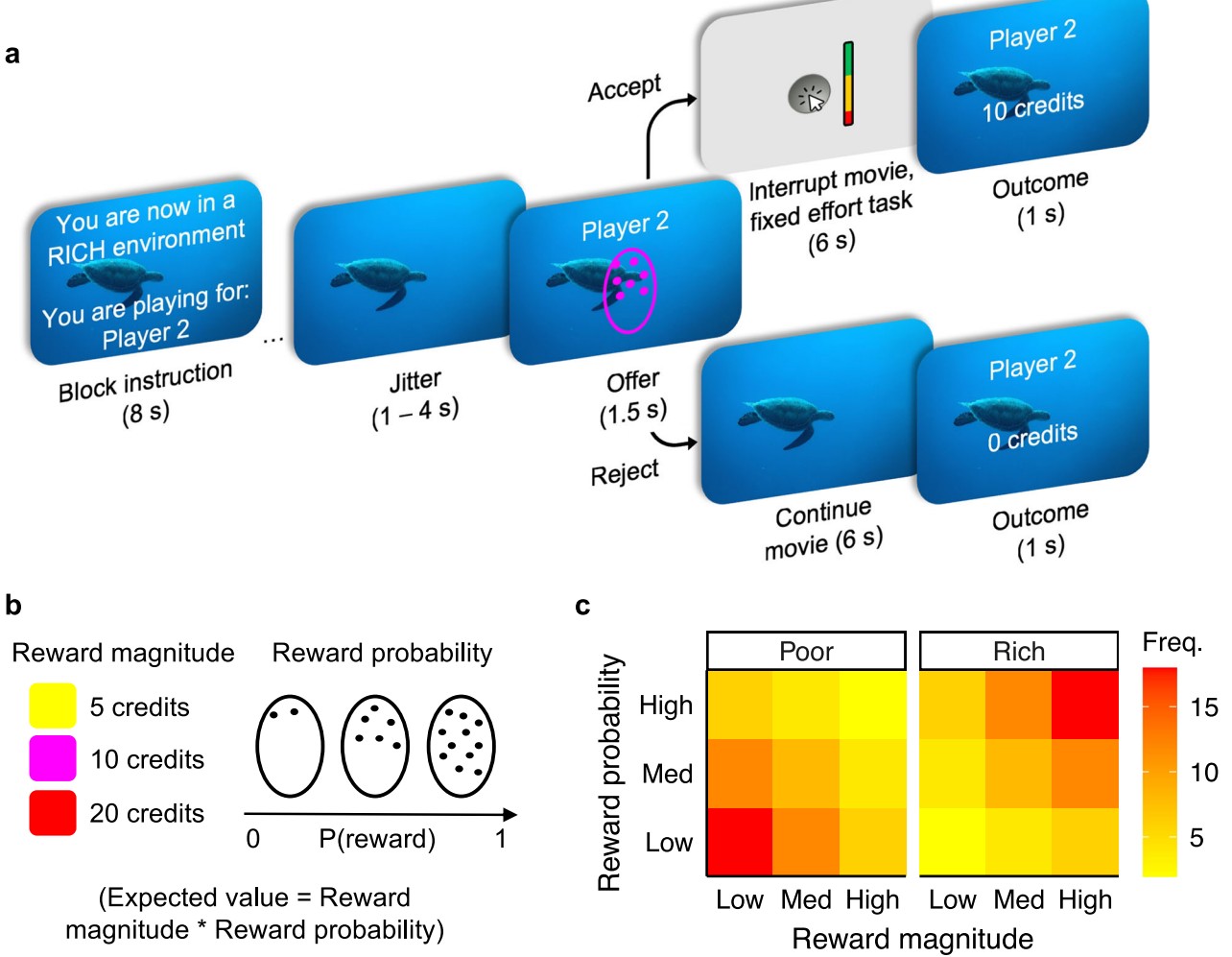

**Fig. 1 | Prosocial ecology task.** Participants watching an episode of a nature documentary encountered opportunities to earn rewards. **a** The task was divided into blocks of trials that differed based on the foraging environment and reward recipient. Each block signalled to the participant whether the environment was poor or rich and who they were playing for (themselves or an anonymous other). Within each block, participants saw opportunities (coloured ovals with dots inside) to earn potential rewards. The colour and number of dots (**b**) represented the potential reward's magnitude and probability, respectively. If participants chose to accept the opportunity, the movie was muted and hidden, and a brief effort task appeared (note: study 3's effort task duration was 3 s). Here, participants needed to repeatedly press an on-screen button (studies 1 and 2, displayed here) or squeeze a handheld grip device (study 3; see "Methods" section) to 60% or 50% of their maximum effort threshold, respectively. The screen would then display the amount of reward earned. If participants instead chose to decline the opportunity, no action was required, and the movie continued to play uninterrupted. **c** Both environments saw the same number and range of reward magnitudes and probabilities. However, on average, reward opportunities in the poor environment were lower in magnitude and probability compared to those in the rich environment. The frequency shown in the heatmap here refers to the number of trials in studies 1 and 2 (study 3 had twice as many trials).

value of the preceding opportunity (OR = 0.97 [0.93, 1.01], $z = 1.66$, $p = 0.097$, $BF_{01} = 46.53$; see Table S2 for full results). In other words, the observed three-way interaction on participants' choices was present even after accounting for potential effects of fatigue or autocorrelation in choices (e.g. by repeating the same actions from previous trials).

We also tested our preregistered analysis of reward magnitude and probability as separate predictors to examine if expected value effects were driven by reward magnitude or probability alone. We observed that participants were more likely to choose to act at higher reward magnitudes (OR = 2.33 [2.09, 2.60], $z = 15.02$, $p < 0.001$) and at higher probabilities (OR = 7.05 [6.00, 8.32], $z = 23.51$, $p < 0.001$). People were more likely to choose to act for themselves than for others when the reward's probability was high (probability*recipient: OR = 0.83 [0.71, 0.97], $z = 2.29$, $p = 0.022$, see Table S3 for full model results).

Second, we ran a control analysis to test whether participants could still exert the required effort (60% of their maximum button presses) at the end of the experiment by measuring participants' effort thresholds again after completing the task (see "Methods" section). We found that people were still able to achieve the required effort, with higher effort post-task compared to the start ($t_{(233)} = 10.55$, $p < 0.001$, $d = 0.69$ [0.60, 0.79]). We also checked whether participants were able to successfully perform the effort task after choosing to act and found that participants were indeed highly successful ($M = 99.4\%$ success rate) for both self and other. To mimic everyday life contexts, it was important that participants found the background task of watching the movie enjoyable. We therefore asked participants at the end of the study how much they enjoyed watching the movie, and whether they had seen it previously. The majority of participants had not seen the movie before (208 *no* vs 29 *yes*, $\chi^2_{(1)} = 135.19$, $p < 0.001$) and overall enjoyment was high ($M (SD) = 6.90$ (2.09), 0 = did not enjoy at all, 9 = very much enjoyed; $t$-test against neutral rating: $t_{(236)} = 17.64$, $p < 0.001$, $d = 1.15$ [0.93, 1.41]). This suggests that interrupting the movie indeed elicited an opportunity cost of missing out on the movie

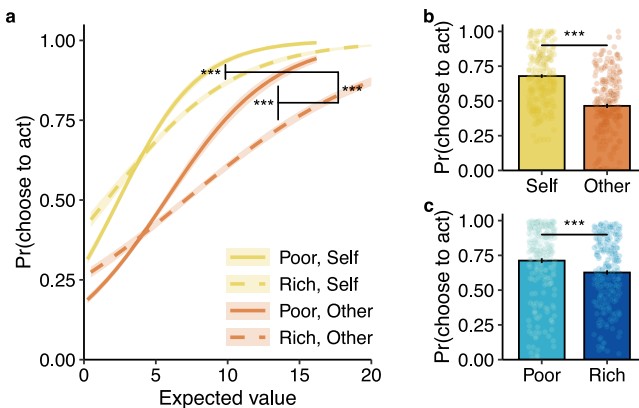

**Fig. 2 | Humans decide to help others more often in poor foraging environments.** Results from study 1 ($n = 237$). **a** There was a significant interaction between recipient, environment, and reward expected value (three-way interaction: OR = 1.49 [1.05, 2.12], $z = 2.20$, $p = 0.028$). As the expected value increased, participants were more likely to act to help others in poor foraging environments compared to rich ones. This environmental effect was less influential on decisions that benefited oneself. The shaded bands around the lines represent 95% confidence intervals of the mean. **b** In line with previous studies, participants chose to exert effort to earn rewards more for themselves than for others (self vs other: OR = 0.14 [0.10, 0.19], $z = 12.94$, $p < 0.001$). **c** Participants were more likely to choose to act in the poor environment compared to the rich one, for equivalent opportunities (poor vs rich: OR = 0.50 [0.40, 0.62], $z = 6.36$, $p < 0.001$). Plotted here are the probabilities of accepting opportunities with the same mean expected value across both environments. Each dot represents a participant's average for the given condition. Error bars represent the standard error of the mean; *** $p < 0.001$; all tests were two-sided Wald $Z$-tests.

to pursue an alternative. We also asked participants at the end of the task how likely they were to respond in the poor and rich environments, and intriguingly, they believed they were more likely to respond in rich environments (see Supplementary Results for full details), suggesting the stronger influence of the poor environment on decisions to act was at least partly implicit. In summary, study 1 showed participants were more likely to decide to act to help others in poor foraging environments, and these effects were not driven by fatigue, the influence of the previous trial, or inability to perform the effort task.

### Opportunities to help are accepted more for self than others
Previous studies have shown that despite humans being willing to help others, for example by splitting amounts of money[21] or putting in effort to earn rewards[23,24,54,55], they are somewhat selfish and much less willing to benefit others than they are to benefit themselves. In such studies, people will unevenly divide a pot of money between themselves and others and exert less effort to win rewards for others. We found that even though participants had the option to carry on watching the movie, they still interrupted their behaviour to help others a significant portion of the time (46.44%, $SD = 23.49\%$; $t$-test comparing against 0%: $t_{(236)} = 30.43$, $p < 0.001$, $d = 1.98$ [1.77, 2.24]). However, they maintained a self-bias consistent with other studies, choosing to act to help others less often compared to opportunities that benefitted themselves (self vs other: OR = 0.14 [0.10, 0.19], $z = 12.94$, $p < 0.001$; Fig. 2b). We also observed that the self vs other difference increased at greater expected values (recipient*expected value: OR = 0.48 [0.39, 0.59], $z = 6.94$, $p < 0.001$).

### People are more willing to act in poor foraging environments
Research in behavioural ecology shows that the richness of the environment affects animals' decisions to forage for food[4,51,52]. Models of prey selection predict that in rich environments, where the quantity

and quality of food opportunities are high, animals will become more selective in their foraging behaviour. In poorer environments, animals become less selective and are more likely to forage for food that they might have otherwise refused in a rich environment. In line with these models, we found that, for equivalent opportunities, people were more likely to choose to act on opportunities in the poor environment compared to the rich one (poor vs rich: OR = 0.50 [0.40, 0.62], $z = 6.36$, $p < 0.001$; Fig. 2; see Table S1 for full results). For example, participants encountered medium magnitude/probability opportunities at the same rate in both environments (see central squares in Fig. 1c; see also Supplemental Results for control analyses on this subset of trials). However, they were significantly more likely to accept an opportunity when it appeared in the poor environment compared to when the same opportunity appeared in the rich environment[9] (see also Supplemental Results).

### Greater environmental effect on money earned for others
Finally, we tested whether differences in decisions to act were reflected in the number of credits participants earned for themselves and the other person, which translated into bonus money at the end of the study. Importantly, the opportunities presented for self were identical to those presented for others, across both environments. We found that participants earned more credits in the rich environment relative to the poor one (poor vs rich: $b = 146.93$ [141.68, 152.19], $z = 54.87$, $p < 0.001$), and more credits for themselves than for others (self vs other: $b = -33.85$ [$-39.11$, $-28.59$], $z = 12.64$, $p < 0.001$). Reflecting choice behaviour, we found a significant interaction between environment and recipient, showing that participants earned more money for others in the poor environment relative to the rich one, compared to this difference for themselves ($b = -35.25$ [$-45.76$, $-24.74$], $z = 6.58$, $p < 0.001$; Fig. S1).

### Reliable environmental influences on helping others
Next, we sought to replicate our effects in an independent online sample of participants ($n = 219$, mean age = 27.7 years (SD = 4.8), 108 women, 107 men, 4 non-binary). For study 2, we preregistered (AsPredicted #107076) the same task and hypotheses as in study 1. We again found a stronger effect of environment on choices to act for others relative to oneself (three-way interaction: OR = 2.55 [1.67, 4.01], $z = 4.21$, $p < 0.001$, Fig. 3a, see Table S4 for full results). This effect again remained significant after accounting for potential effects of fatigue and previously encountered opportunities (three-way interaction: OR = 2.51 [1.65, 3.92], $z = 4.18$, $p < 0.001$). As in study 1, participants were less likely to choose to act as the task progressed but were more likely to act if they had done so on the preceding opportunity (all $p$s < 0.015). Here, participants were also less likely to act if the preceding expected value was higher ($p = 0.017$; see Table S5 for full results). Again, participants reported enjoying watching the movie ($M$ (SD) = 6.89 (2.05), $t$-test against neutral rating: $t_{(218)} = 17.27$, $p < 0.001$, $d = 1.17$ [0.94, 1.45]) and the majority and not seen it before (175 no vs 44 yes, $\chi^2_{(1)} = 78.36$, $p < 0.001$). All other key findings from study 1 replicated (see Supplementary Results).

### Environmental effects extend to different types of effort and recipients
In study 3, we tested whether our findings extended to different forms of effort and recipients ($n = 54$, aged 18–32 years, $M$ (SD) = 21.9 (3.2), 39 women, 15 men). We invited participants to the lab, where they anonymously met the other person before performing the same task as above. To introduce the participant to the other person, we used a previously established role assignment procedure[23,24,56] (see "Methods" section). This procedure allows participants to meet each other, yet does not reveal any identifying information such as gender, age, or physical characteristics to minimise any influence of these factors on prosocial decisions. For this in-person version, participants used a

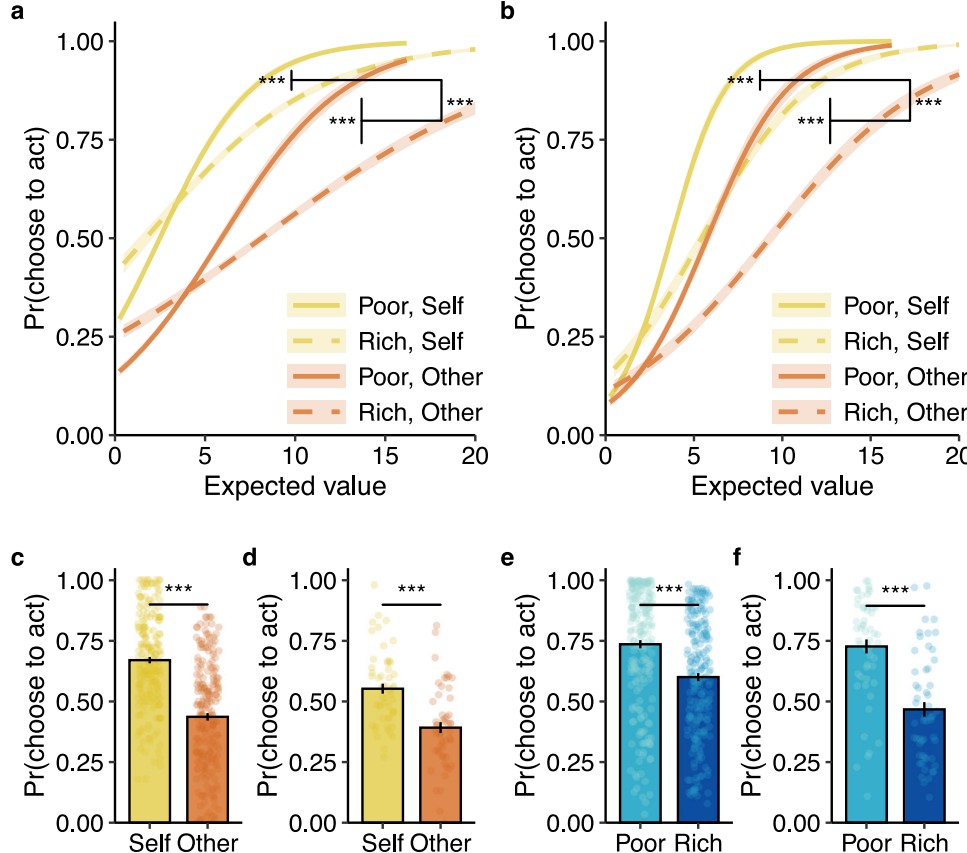

**Fig. 3 | Environmental effects on deciding to help others are consistent across studies.** Results from study 2 ($n = 219$) and study 3 ($n = 54$). For all three studies, there was a significant interaction between recipient, environment, and expected value. **a**, **b** (studies 2 and 3) At higher expected values, participants were more likely to act to help others in poor foraging environments compared to rich ones (study 2: three-way interaction OR = 2.55 [1.67, 4.01], $z = 4.21$, $p < 0.001$; study 3: three-way interaction OR = 1.92 [1.32, 2.80], $z = 3.40$, $p < 0.001$). When the decision to act benefited oneself, the environment had less of an impact. The shaded bands around the lines represent 95% confidence intervals of the mean. **c**, **d** (studies 2 and 3) On average, participants chose to act to earn rewards more for themselves than for

others (study 2: self vs other: OR = 0.10 [0.07, 0.13], $z = 13.52$, $p < 0.001$; study 3: self vs other: OR = 0.20 [0.13, 0.31], $z = 7.72$, $p < 0.001$). **e**, **f** (studies 2 and 3) Participants were more likely to choose to act for a given opportunity when it appeared in a poor environment as compared to a rich one (study 2: poor vs rich: OR = 0.29 [0.22, 0.37], $z = 9.59$, $p < 0.001$; study 3: poor vs rich: OR = 0.24 [0.19, 0.30], $z = 12.56$, $p < 0.001$). Plotted here are the probabilities of accepting opportunities with the same mean expected value across both environments. Each dot represents a participant's average for that condition. Error bars represent the standard error of the mean; ***$p < 0.001$; all tests were two-sided Wald $Z$-tests.

grip-force device to exert physical effort, thresholded to their own ability.

Replicating our findings from studies 1 and 2, we found a stronger effect of the environment on participants' decisions of when to act to help others (three-way interaction: OR = 1.92 [1.32, 2.80], $z = 3.40$, $p < 0.001$; Fig. 3d, see Table S6 for model results). This significant interaction remained when controlling for trial number, as well as the choice and expected value on the preceding opportunity (three-way interaction: OR = 2.05 [1.40, 3.01], $z = 3.68$, $p < 0.001$). Participants were again less likely to act on an opportunity over time and more when the preceding expected value was higher (all $p$s < 0.001), but we found no significant effect of their previous choice on their current decision to act ($p = 0.379$; see Table S7 for model results). Participants again enjoyed watching the movie ($M$ ($SD$) = 7.59 (1.47), $t$-test against neutral rating: $t_{(53)} = 15.51$, $p < 0.001$, $d = 2.11$ [1.59, 3.05]) and the majority had not seen it before (45 *no* vs 9 *yes*, $\chi^2_{(1)} = 24.00$, $p < 0.001$). All other key findings remained the same as in study 1 and study 2 (see Supplementary Results).

### Participants over-exert themselves in a poor environment
Study 3 additionally allowed us to examine how much physical force participants exerted in the different conditions. We calculated the normalised area under the curve of the force exerted and examined

whether this was modulated by recipient, environment, and expected value (see "Methods" section). Participants exerted more force at higher expected values in the poor environment compared to the rich one (environment*expected value: $b = -0.016$, [$-0.026$, $-0.005$], $z = 2.94$, $p = 0.003$; Fig. 4), but there was no overall interaction between recipient and environment ($b = 0.004$ [$-0.022$, $0.029$], $z = 0.29$, $p = 0.776$, BF$_{01}$ = 82.29). We found that participants exerted more force to benefit themselves than for others overall (self vs other $b = -0.054$ [$-0.074$, $-0.034$], $z = 5.52$, $p < 0.001$). Together, these findings suggest that participants over-exerted in poor foraging environments when the expected value was higher and on opportunities for themselves.

### Opportunity cost parameters are encoded distinctly for prosocial and self-benefiting decisions
Mixed-effects models showed that participants were more influenced in poor foraging environments than in rich ones to help others compared to themselves. However, these models cannot quantify latent influences on participants' behaviour in terms of weighting opportunity costs, sensitivity to value, or non-linear influences on decision-making. We therefore built a wide range of computational models based on decision neuroscience to quantify how opportunity costs of the different environments influenced decisions to act for self and other, and fit them to the choice data from the 3 studies to maximise

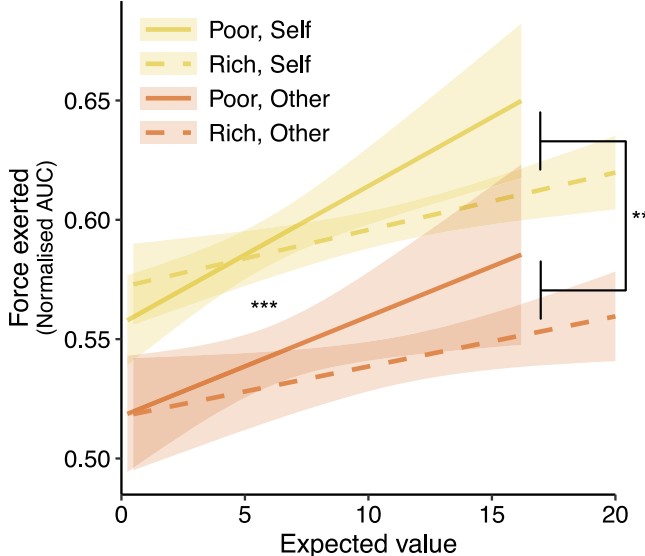

**Fig. 4 | Participants over-exert to earn rewards for themselves and in poorer foraging environments.** Overall, participants in study 3 ($n = 54$) exerted significantly more physical force when trying to win potential rewards for themselves than for others ($b = -0.054$ [$-0.074$, $-0.034$], $z = 5.52$, $p < 0.001$). There was a significant interaction between environment and expected value, which showed that participants tended to exert more force in the poor environments when the expected value was higher ($b = -0.016$, [$-0.026$, $-0.005$], $z = 2.94$, $p = 0.003$). AUC area under the curve; shaded bands around the lines represent 95% confidence intervals of the mean; $^{**}p < 0.01$, $^{***}p < 0.001$; all tests were two-sided Wald Z-tests.

power (total $n = 510$). These models quantified whether opportunity costs ($o$ parameter) differed between environments and recipients, or whether a single parameter applied across all conditions. Higher opportunity costs suggest that pursuing the current opportunity may be less worthwhile than alternatives (other reward opportunities, continuing to watch the movie, avoiding the effort task, etc.). We also included a value sensitivity (inverse temperature) parameter ($\beta$) that captured the degree to which value sensitivity influenced choice consistency and to fit the models to behaviour. Each class of model tested this parameter between environments and recipients, or as a single parameter across conditions. Finally, based on studies in economics, we included models that varied the functional form by which opportunity costs discounted probabilistic rewards[57–61] (see "Methods" section).

We found that participants' choices to act were best explained by a model with distinct opportunity costs ($o_{self/poor}$, $o_{self/rich}$, $o_{other/poor}$, $o_{other/rich}$) and value sensitivity ($\beta_{self/poor}$, $\beta_{self/rich}$, $\beta_{other/poor}$, $\beta_{other/rich}$) parameters for each environment and recipient, as well as a single risk aversion ($\alpha$) parameter on the reward magnitude. This model had the highest exceedance probability (95.8%) and explained a large portion of the variance in choices ($R^2 = 0.83$). Data simulations showed that the model was robust and had excellent model identifiability (Fig. 5a) and parameter recovery (Fig. 5b, see "Methods" section). Fitting the models to the data from each study separately also revealed separate $o$ parameters for each environment and recipient, as well as a risk aversion parameter on reward magnitude (see Tables S8–S10).

First, comparing the four opportunity cost parameters, we found a larger environmental effect on opportunity costs to help others compared to helping oneself (recipient*environment: $b = 0.10$ [0.05, 0.15], $z = 4.27$, $p < 0.001$; Fig. 5c), reflecting the model-free results. Opportunity costs were lower in poor environments (poor vs rich: $b = 0.13$ [0.11, 0.15], $z = 10.83$, $p < 0.001$) and when the outcome affected oneself (self vs other: $b = 0.34$ [0.32, 0.37], $z = 28.82$, $p < 0.001$). In other words, opportunity costs were lower in poor foraging

environments compared to the rich ones, and when the decision benefited another person this difference between environments widened.

Next, we found that value sensitivity to other-benefitting opportunities in the poor environment reached the same level as in the rich environment for oneself. A Bayesian $t$-test provided strong Bayesian evidence of no difference between the two conditions ($t_{(509)} = 1.22$, $p = 0.227$, $BF_{01} = 19.32$, Fig. 5d), suggesting that participants were equally likely to follow their preferences for others in the poor environments as for self in the rich ones. We additionally observed that participants were more value-sensitive to opportunities that benefitted themselves overall (self vs other: $b = -1.48$ [$-1.66$, $-1.31$], $z = 16.47$, $p < 0.001$) and to those that appeared in poor environments (poor vs rich: $b = -1.33$ [$-1.51$, $-1.16$], $z = 14.79$, $p < 0.001$). The interaction between recipient and environment was not statistically significant ($b = 0.34$ [$-0.02$, 0.69], $z = 1.87$, $p = 0.062$). Finally, we found that participants were generally risk averse at higher reward magnitudes (one-sample $t$-test on the $\alpha$ parameter: $M$ ($SE$) = 0.39 (0.01); $t_{(509)} = 60.71$, $p < 0.001$, $d = 2.69$ [2.43, 3.00]).

## Opportunity costs for other-benefitting decisions are related to empathy and utilitarianism

We next examined the relationship between the estimated computational parameters and individual differences in social cognition and behaviour, as well as depression, anxiety, and apathy. To do so, we conducted a factor analysis on the questionnaires that participants completed (see "Methods" section). This allowed us to extract underlying dimensions of behaviour measured across the questionnaires (e.g. depression and anxiety scores were highly correlated, $r_{(507)} = 0.77$) while also aiding in conceptual interpretation and statistical inference by reducing the number of comparisons. The analysis revealed three distinct dimensions across the subscales of the questionnaires (Fig. 6a). Factor 1 ('Psychiatric traits') included high loadings from measures of depression, anxiety, and behavioural and social apathy, Factor 2 ('Empathy and emotional motivation') included high loadings from measures of cognitive and affective empathy and emotional apathy, and Factor 3 ('Utilitarianism') included high loadings from measures of utilitarianism—i.e. beliefs related to maximizing well-being for all.

We correlated each factor with participants' parameter estimates from the winning model (Fig. 6b; $p$-values were FDR adjusted, see "Methods" section). The factor 'Utilitarianism' showed significant negative correlations with opportunity costs in both the rich ($r_{(506)} = -0.18$, $p < 0.001$) and poor ($r_{(506)} = -0.14$, $p = 0.007$) environments when deciding for others, but not when deciding for oneself (self-poor: $r_{(506)} = -0.06$, $p = 0.293$, $BF_{01} = 8.05$; self-rich: $r_{(506)} = -0.01$, $p = 0.847$, $BF_{01} = 17.54$). The factor 'Empathy and emotional motivation' also showed significant negative correlations with opportunity costs when deciding for others in both the rich ($r_{(506)} = -0.11$, $p = 0.034$) and poor environments ($r_{(506)} = -0.10$, $p = 0.046$), but not when deciding for oneself (self-poor: $r_{(506)} = -0.03$, $p = 0.585$, $BF_{01} = 14.35$; self-rich: $r_{(506)} = -0.01$, $p = 0.860$, $BF_{01} = 17.55$). This suggests that, for those higher in empathy and with stronger beliefs about maximising others' well-being, the opportunity costs when deciding to help others are lower, leading them to be more likely to interrupt their ongoing behaviour to help. Importantly, there was also Bayesian evidence that psychiatric traits did not explain variance in computing opportunity costs for others (all $BF_{01}s > 3.70$; see Table S11).

## Discussion

Prosocial behaviours are key to building and maintaining social connections and well-being. Across 3 studies, we show that deciding when to help others is indeed influenced by one's environment. Participants were more likely to act to earn rewards in poor compared to rich foraging environments, and the strength of this environmental effect

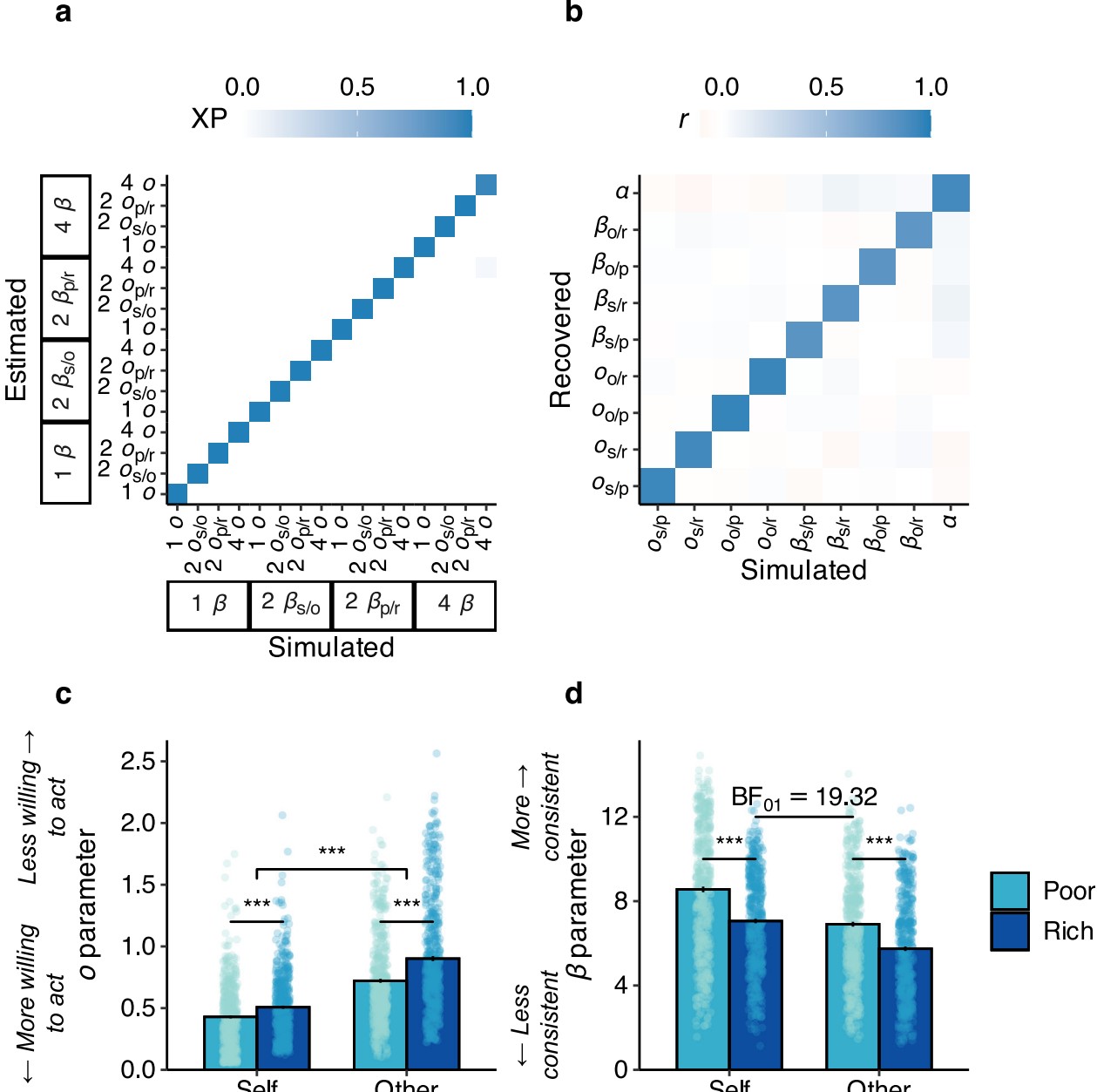

**Fig. 5 | Computational modelling revealed that the opportunity costs of different foraging environments were encoded distinctly for others and the self.** The winning model included separate opportunity cost and value sensitivity parameters for self and other in each environment, as well as a risk aversion parameter on the potential reward's magnitude ($n = 510$). **a** Each model was robustly identifiable compared to other models in the model space. Presented here are the exceedance probabilities for simulated vs estimated models that included a risk aversion parameter on reward magnitude (see Fig. S2 for fit statistics for all models). **b** Simulations of the winning model showed good recovery of all parameters (Pearson's $r$s > 0.80). **c** The winning model showed a significant interaction between recipient and environment where opportunity costs were lower when deciding to help others in poor compared to rich foraging environments (interaction: $b = 0.10$ [0.05, 0.15], $z = 4.27$, $p < 0.001$; poor vs rich: $b = 0.13$ [0.11, 0.15],

$z = 10.83$, $p < 0.001$; self vs other: $b = 0.34$ [0.32, 0.37], $z = 28.82$, $p < 0.001$). **d** There was strong Bayesian evidence of no difference in value sensitivity to opportunity costs of prosocial decisions in the poor environments compared to self-benefiting decisions in the rich environments (Bayesian $t$-test: $t_{(509)} = 1.22$, $p = 0.227$, $BF_{01} = 19.32$). Participants were more value-sensitive to opportunities that benefitted themselves overall (self vs other: $b = -1.48$ [−1.66, −1.31], $z = 16.47$, $p < 0.001$) and to those that appeared in poor environments (poor vs rich: $b = -1.33$ [−1.51, −1.16], $z = 14.79$, $p < 0.001$). Error bars represent the standard error of the mean; the coloured dots represent the estimated parameters for each participant. The axis label subscripts s, o, r, and p in (**a**, **b**) represent the self, other, rich, and poor conditions, respectively. XP = exceedance probability; ***$p < 0.001$; all tests were two-sided Wald $Z$-tests.

was enhanced when deciding for others. Computational modelling revealed that decisions were determined by comparing expected rewards with different opportunity costs for each environment and recipient. Distinct factors of utilitarianism, and empathy/motivation, but not psychiatric traits, were related to variability in computing opportunity costs when deciding for others. People were also similarly

sensitive to the value of opportunities in the rich environments for themselves as for opportunities in the poor environments for others. Together, these findings show a stronger foraging environmental influence on decisions about when to help others compared to decisions for ourselves, highlighting the fundamentally context-dependent nature of human prosocial behaviour[13].

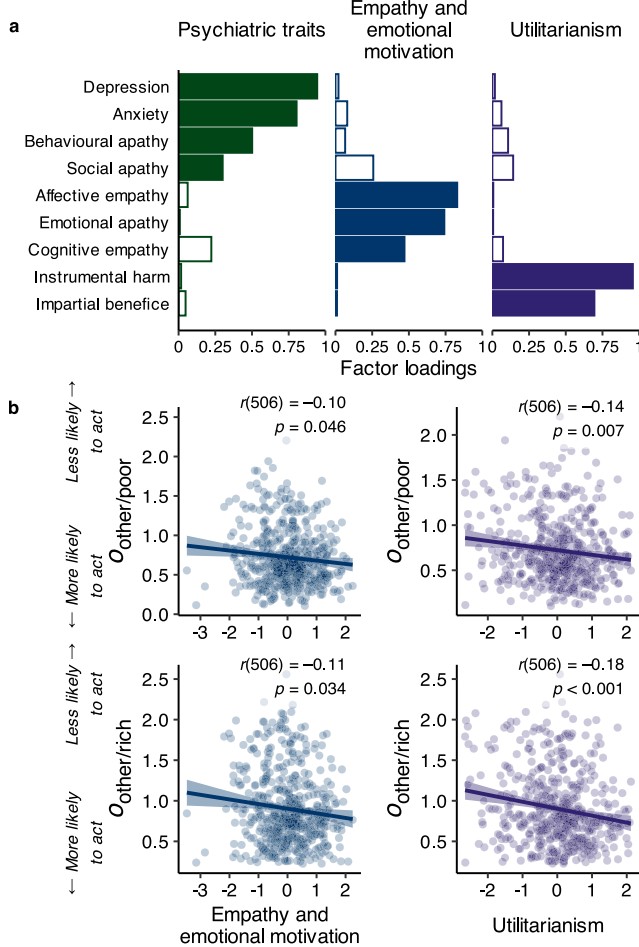

**Fig. 6 | Variability in opportunity costs for others relates to empathy and utilitarianism. a** Individual factor loadings for the subscales of the questionnaires administered (*n* = 508). Our factor analysis revealed three dimensions of behaviour related to psychiatric traits, empathy and emotional motivation, and utilitarianism. Note: the loadings displayed here are absolute values; coloured bars represent loadings > 0.30. **b** The factors 'Empathy and emotional motivation' and 'Utilitarianism' are related to opportunity costs when deciding for others in both rich and poor foraging environments. Those who are more empathetic and emotionally motivated, or with stronger utilitarian beliefs, may find helping others less costly and may be more willing to interrupt their behaviour to help others. Error bands represent the standard error of the mean; *r* values represent Pearson's correlation coefficient; *p*-values are two-sided and FDR-adjusted.

In the laboratory, prosocial behaviours are often assessed using economic games, where two or more options are presented simultaneously, and often a decision is required. This can give rise to behaviour that appears prosocial, but may instead be a consequence of the study's design—e.g. choosing an effortful option to avoid boredom or to appease the experimenter[62–64]. In everyday life, however, choosing between options is rarely done simultaneously[1,2,65,66]. Rather, choices are made sequentially, where the decision is either to accept or reject the current option[67]. For example, when searching for a charity to donate to, we often accept or reject donating to options as we encounter them, rather than seeing two or more options simultaneously. Our findings suggest that prosociality is strongly influenced by the context and relative value of opportunities, such as when deciding to donate, in line with established models of animal foraging behaviours. Moving forward, these context-dependent manipulations can be extended to examine key individual differences in human prosocial behaviour, such as across the lifespan.

A central debate in social psychology is whether people with lower income or financial well-being are more or less prosocial. Existing work has often used self-report or observational designs, with mixed findings about whether or how environments alter prosociality[17,44–49]. Here, using a prey-selection paradigm adapted from behavioural ecology, we show that directly manipulating one's environment to be poor can shift decisions towards helping others. This finding may at first appear counterintuitive, as opportunities in the poor environment were overall worse compared to the rich environment. However, this pattern of behaviour can be seen in many non-human animals when foraging for food[4,6,25]. For animals in rich environments, where high-quality food is abundant, it is advantageous to reject low-quality food and instead wait for better options. In poor environments, where high-quality food is scarce, waiting for better options may be too costly. Instead, the animal should be less selective and feed on lower-quality prey that it may have otherwise ignored. This suggests the environmental effect we observed may reflect fundamental foraging mechanisms shared with other non-human animals[1,3,7,12], and which may be enhanced for other-benefitting decisions.

An interesting avenue for future research is to observe these mechanisms in more natural settings by manipulating one's immediate, real-world environment[68,69]. For example, deciding whether to donate to a charity whose impact is smaller in scale, although still important, may depend on the abundance and relative value of other charities encountered (i.e. the environment's richness). How these environmental factors translate to one's broader socioeconomic environment remains to be investigated. Different social psychological theories could inform and be informed by our findings. For example, scarcity theory proposes that being in a poor environment induces a scarcity mindset[70], which results in suboptimal decisions and behaviours. Here, we suggest that poorer foraging environments encourage people to act to help others, whether this behaviour can be viewed as optimal is a question for future studies. Other theories suggest that individuals in poor environments may be more interdependent and place a higher value on social ties that predispose them toward helping. However, the current results demonstrate that people can be moved towards more or less helpful behaviour simply and directly as a function of the richness of opportunities in one's environment, irrespective of the importance of social ties. Future studies should examine whether these environmental differences are modulated by participants' pre-existing financial situation, for example, by measuring financial well-being or endowing participants with monetary amounts before the experiment.

Our computational modelling suggests that the contextual features of one's environment are used to decide when to act. Previous work in humans and non-human primates suggests that these features are tracked via a habenula–insular circuit[11,71]. Future work should examine how the brain integrates these features with opportunity costs, and whether, like for physical effort, they are represented in distinct areas for self and others[24]. Opportunity costs are also related to individual levels of empathy/emotional motivation and utilitarianism, yet not to psychiatric traits. This suggests that people who are more empathetic and emotionally motivated, or with stronger utilitarian beliefs, find it less costly and are more willing to interrupt their behaviour to help others, which may challenge purely cognitive accounts of vicarious experience as a motivator for decisions to act. It also supports theoretical accounts of the links between empathy and altruism[72], showing that in paradigms where people engage in costly helping, such costly helping is more prevalent in individuals with higher self-reported empathy. Intriguingly, emotional motivation is also crucial here[73,74], with those reporting the highest emotional motivation weighting opportunity costs less strongly and therefore being more willing to act to help others.

Our computational modelling also allowed us to compare sensitivity to the rewards for self and other, and between rich and poor

foraging environments. The value sensitivity (i.e. inverse temperature) parameter captures the stochasticity in choices. Higher values indicate that choices are more strongly influenced by an opportunity's value, and lower values indicate more randomness or less influence on choices. Intriguingly, we found that people, on average, were similarly value sensitive to offers for themselves in the rich environments as for others in poor ones, suggesting that self and other became more computationally similar depending on the environmental context in which the decision is being made. Together with our choice behaviour findings, people appear to be only slightly selfish in poor foraging environments but act more selfishly in rich environments. This finding is particularly interesting as the majority of studies testing how we decide for others in the context of effort and rewards suggest that we are less value sensitive for others compared to ourselves[23,24,75]. Future studies could probe whether corresponding neural representations follow this same pattern, particularly in brain areas such as the anterior cingulate gyrus and subgenual anterior cingulate cortex, which have been shown to track prosocial decisions more strongly than self-benefiting ones[76].

In addition to these strengths, our study also has limitations. Moment-to-moment changes in the salience of the movie may have affected participants' decisions about whether to act. We showed 4 different episodes of the nature documentary across participants, counterbalanced block orders and timings, and across all three studies, found that most participants enjoyed watching the movie and had never seen it before. This suggests that their decisions to interrupt the movie to act were biased by true opportunity costs, rather than simply a desire to avoid a boring movie. Moreover, by including the movie as a baseline option rather than a plain screen, which is common in psychology experiments, we could measure behaviours in a more ecological context and try to rule out effects of boredom or demand characteristics of doing something on decisions to act. Relatedly, it would be interesting to examine whether and in what species the environmental influence on helping conspecifics is present. The extent of prosociality seen in non-human animals is a widely debated question[77–81] and our paradigm could be adapted to examine the boundaries of the prosocial environment effect across species. We focused on physical effort as the additional cost participants had to exert to help themselves or others. Testing how the environmental effect extends to different types of cost, such as cognitive or time costs, such as stopping our own work to help a colleague, is important for future work.

Overall, we show that humans are more likely to choose to help others in poor than in rich foraging environments. Variability in tracking opportunity costs of different environments for others relates to empathic and utilitarian traits but not to anxiety or depression. Moreover, the sensitivity to values that help others in poor environments is similar to helping oneself in a rich environment. These findings reveal a fundamental but overlooked aspect that drives variability in human prosociality with implications for elucidating when and why humans decide to help others. The boundaries of this environmental influence could be critical for understanding prosocial decision-making as environments change.

## Methods

### Participants
Studies 1 and 2 were approved by the research ethics committee at the University of Birmingham; study 3 was approved by the research ethics committees at the University of Birmingham and the University of Oxford. All participants provided written informed consent. Online participants were compensated at the rate of £6 per hour with a potential bonus of up to £2; in-person participants were compensated at the rate of £10 per hour with a potential bonus up to £5.

**Study 1.** We recruited 323 people through the online platform Prolific to take part in the study. Data were collected between June 2022 and

August 2022. Our preregistered exclusion criteria (AsPredicted #102887, 20 July 2022, https://aspredicted.org/nfmd-m9r7.pdf) included failing at least 50% of attention checks, and overall acceptance rates below 10% in the prosocial ecology task. In total, 14 participants were excluded for failing the attention checks. We also excluded participants from analysis based on the following criteria that we did not preregister: failing an attention check question in the Questionnaire of Cognitive and Affective Therapy (QCAE; 6 participants), failing to correctly self-identify as Player 1 in debriefing questions at the end of the session (12 participants), and those with acceptance rates above 90% (54 participants). This last criterion was used in an earlier study with a similar design[11]. If a participant met one or more of the above criteria, they were excluded from the final sample and further analysis. The final sample consisted of 237 people (aged 18–35 years, M (SD) = 28.8 (4.4); self-reported gender: 124 women, 112 men, 1 non-binary).

**Study 2.** We recruited 301 people through the online platform Prolific to take part in the study. Data were collected between September 2022 and October 2022. We preregistered the same exclusion criteria as study 1 in addition to excluding participants with overall acceptance rates above 90% (AsPredicted #107076, 15 Sept. 2022, https://aspredicted.org/3ctx-7r9n.pdf). In total, 16 participants failed at least 50% of the attention checks, 1 participant had an overall acceptance rate below 10%, and 46 participants had overall acceptance rates above 90%. We also excluded participants who failed the QCAE attention check (10 participants) and those who failed to self-identify as Player 1 in the debriefing questions (9 participants). The final sample consisted of 219 people (aged 18–35 years, M (SD) = 27.7 (4.8), self-reported gender: 108 women, 107 men, 4 non-binary).

**Study 3.** We recruited 55 people through the University of Birmingham and University of Oxford communities to participate in the study; this study was not preregistered. Data were collected in two phases: between November 2019 and March 2020 and between December 2022 and January 2023. We used the same exclusion criteria as in studies 1 and 2. Study 3, however, did not include attention checks within the task, as the experimenter was present in the room to ensure that participants were attending to the task. One participant had an acceptance rate below 10%; this participant also failed the QCAE attention check. In total, 54 participants were included in the final sample (aged 18–32 years, M (SD) = 21.9 (3.2), self-reported gender: 39 women, 15 men).

### Prosocial ecology task
Participants watched an episode of the nature documentary *Blue Planet II* (BBC Studios, 2017) during which opportunities to earn rewards appeared on the screen (Fig. 1). Opportunities were presented as coloured ovals with dots inside. The colour of the oval and dots represented the magnitude of the potential reward (5, 10, or 20 credits) and the number of dots represented the probability of being given the reward (0–100%; Fig. 1b). The colour–magnitude combinations were randomised between participants. If the participant chose to accept the opportunity, they did so by pressing the space bar on the computer's keyboard. Doing so would mute and hide the movie while an effort task appeared on screen. Crucially, the movie continued to play while hidden, so deciding to act on the opportunity meant that the participant would miss part of the movie. For the effort task in studies 1 and 2, participants had 6 s to rapidly press an on-screen button until they reached 60% of their maximum number of presses (see Procedure section). For study 3, participants needed to squeeze and maintain a handheld dynamometer to at least 50% of their maximum voluntary contraction (MVC; see "Procedure" section). If participants failed to exert enough effort within 3 s, they failed the effort task and did not receive any reward. If participants exerted enough effort, they were

given the reward based on its probability (i.e. the number of dots in the opportunity).

The task was divided into 4 separate blocks (8 blocks in study 3) that differed in foraging environments and the recipient of the rewards (each unique block was presented twice in study 3). In half of the blocks, participants could earn rewards for themselves and in the other half, they could earn rewards for an anonymous other person. Additionally, in half of the blocks, opportunities appeared in a poor environment and in the other half, they appeared in a rich environment. Importantly, both environments saw the same types and range of opportunities. For example, high magnitude/high probability rewards appeared in the poor environment, and low magnitude/low probability rewards appeared in the rich environment. The average frequency of these opportunities differed between environments (Fig. 1c) such that, in poor environments, the expected value of the reward (magnitude*probability) was, on average, lower compared to the rich environment.

At the start of each block, the task displayed the type of environment and who would receive the rewards. Each block consisted of 36 trials ('opportunities') to earn rewards. Block orders were randomised between participants, and trial orders were pseudorandomized. When an opportunity appeared, participants had 1.5 s to act upon it. If they accepted the offer, the movie was hidden and muted while the effort task appeared; otherwise, the movie continued to play. After the effort task the movie returned, and the screen displayed the number of credits earned. The screen would display "0 credits" based on the reward's probability (more likely if the number of dots was low), if the participant rejected the offer, or if the participant failed the effort task. The duration the movie played was the same whether participants accepted the opportunity or not. Finally, between trials, there was a 1–4 s delay before the next opportunity appeared. We used 4 different episodes of the nature documentary that were counterbalanced across participants. For studies 1 and 2, there were a total of 4 attention checks throughout the task. For this, participants simply needed to press an on-screen button once within 3 s to pass the check. At the end of the session, bonus money was awarded based on the total number of credits earned during all 'self' trials. We did not inform participants of the rate of conversion to ensure that they did not track a running total, but simply that the more credits they earned, the more bonus money they would receive at the end. Online participant data were collected using Gorilla Experiment Builder (www.gorilla.sc). In-person participant data were collected using the Psychophysics Toolbox (version 3.0.11) for MATLAB (version 2012b).

### Questionnaire measures

After completing the main task, participants completed a series of questionnaires and answered questions related to their experience in the task. We asked participants to rate how much they enjoyed watching the movie from 0 (*Not at all*) to 9 (*Very much*); they also indicated whether they had seen the movie before.

**Apathy motivation index (AMI).** We measured individual differences in apathy and motivation using the AMI[82]. The AMI is an 18-item scale where participants are asked to indicate their agreement with each item using a 5-point Likert scale. The scale measures apathy in behavioural, social, and emotional domains as separate subscales. Items are scored such that higher scores indicate higher levels of apathy.

**Questionnaire for cognitive and affective empathy (QCAE).** We measured individual levels of empathy using the QCAE[83]. The QCAE is a 31-item scale that measures cognitive and affective empathy as separate subscales. Participants are asked to rate their agreement with each item using a 4-point Likert scale. For each item, higher scores indicate higher levels of empathy in the given domain.

**Depression and anxiety stress scales (DASS).** We used the shortened version of the DASS to measure individual differences in anxiety and depression[84]. The shortened DASS is a 21-item scale that measures stress, depression, and anxiety using different subscales. Participants are asked to rate how each item applies to themselves using a 4-point Likert scale. For each item, higher scores indicate higher levels of the given emotional state. For the present studies, we excluded questions from the stress subscale, as we were specifically interested in the effects of depression and anxiety.

**Oxford utilitarianism scale (OUS).** We measured individual differences in utilitarian beliefs using the OUS[85]. The OUS is a 9-item scale that measures utilitarian beliefs in positive and negative dimensions using two separate subscales. Participants are asked to rate their level of agreement with each item using a 7-point Likert scale. For each item, higher scores indicate stronger utilitarian beliefs in the given dimension.

### Procedure

**Studies 1 and 2.** Before completing the main task and before any instructions, participants completed an effort thresholding procedure. For this, participants repeatedly pressed an on-screen button as fast as possible for 6 s using the computer mouse. Each button press filled a coloured bar on screen to provide visual feedback. Participants repeated this procedure twice more and were encouraged to fill the coloured bar to a line that was 110% of their current maximum. This thresholding procedure was used to determine how much effort (i.e. number of button presses) was needed in the main prosocial ecology task after accepting an opportunity. Participants then received instructions on how to complete the main task and were told that they would be playing alongside another anonymous person online (Player 2), whose identity would not be revealed to them. In reality, this second player did not exist, and participants simply completed the task on their own. We told participants that on some trials, they could sometimes earn credits that would be given to the other player, that this player would not be earning credits for the participant themselves, and that any credits earned for this other player would not increase the participant's own bonus money at the end of the study. To reinforce the idea that Player 2 was a real person, we introduced them using a fake Prolific participant identifying number that was similar in resemblance to the participant's own number. No participants indicated disbelief that Player 2 was a real person in our debriefing questions at the end of the study. Participants completed 5 practice trials before performing the main task (see Supplementary Material for instructions given to participants). After completing the main task, we measured participants' effort thresholds again using the same procedure from the beginning of the study. Finally, participants completed the questionnaires and answered questions related to the task.

**Study 3.** The procedure used in study 3 was nearly identical to that used in studies 1 and 2. The two key differences were as follows: (1) the modality of the effort task after accepting an opportunity to act, (2) how Player 2 was introduced. For the effort task, participants exerted force by squeezing the handle of a dynamometer rather than pressing an on-screen button. In the main task, participants needed to exert force to at least 50% of their maximum voluntary contraction (MVC; determined using the same effort thresholding procedure as above but instructing participants to squeeze as hard as they can) for a minimum of 1 second to succeed. This effort target was reduced relative to the online version of the task, as squeezing the grip-force device required more effort than repeatedly pressing a computer mouse button. To introduce Player 2, we used a role assignment procedure from earlier social decision-making studies to minimise social preferences of reciprocity[23,24,56]. Participants were anonymously introduced to the other participant, who was a confederate, in person. To ensure that

they remained anonymous, we instructed both participants to remain silent and wear gloves to hide their physical characteristics when meeting. The real participant stood on one side of a door while a second experimenter instructed the confederate to stand on the other side. Both participants were instructed to wave to each other and acknowledge that they had seen the gloved hand of the other. To assign roles to each participant, the experimenter tossed a coin to determine who would pick a ball out of a box first. The colour of the ball determined who was Player 1 and who was Player 2. The procedure was fixed so that the real participant was always assigned the role of Player 1 and the confederate participant assigned the role of Player 2. After the roles were assigned, we emphasised to the real participant that Player 2 would be unaware of Player 1's tasks being performed, and that any rewards earned on their behalf would be anonymous. No participants indicated disbelief that Player 2 was a real person in our questions at the end of the study.

## Statistical analysis

We analysed the behavioural data and fitted computational model parameters using R (v4.3.1). We used the 'glmmTMB' package (v1.17) to fit generalised linear mixed-effects models (GLMMs) to the data and fitted parameters; see Supplementary Methods for full details of the models. Our preregistration stated that we would use the 'lme4' package to fit GLMMs; instead, we chose to use glmmTMB for its similarity to lme4 statistically, yet the implementation of a higher speed of processing for larger datasets such as ours[86]. Model estimates are supplemented with 95% confidence intervals (using the 'uniroot' function), and all significance tests (Wald $Z$-tests) were two-tailed with a threshold of 0.05. Statistical test assumptions were formally tested and visually inspected using the 'DHARMa' package (v0.4.7). Paired $t$-tests were used to compare effort thresholds and are supplemented with Cohen's $d$ effect sizes. Three participants in study 1, and one participant in study 2, did not complete the post-test thresholding procedure and were excluded from this comparison. The 'BayesFactor' package (v0.9.12-4.4) was used to calculate Bayes factors ($BF_{01}$) for non-significant comparisons using the "ultrawide" setting for the priors. For the continuous variables from the GLMMs (e.g. previous expected value), we calculated Bayes factors based on BIC scores[87]. Traditionally, a $BF_{01}$ larger than 3 (equivalent to a $BF_{10}$ smaller than 1/3) is considered substantial evidence in favour of the null hypothesis, with the strength of the evidence increasing at higher BF values[88].

## Computational modelling

We quantified the opportunity costs ($o$) and value sensitivity ($\beta$) for each recipient and environment by comparing multiple discounting models that differed in their shape and number of unique parameters. For all models, we tested for both $o$ and $\beta$ whether a single parameter applied across both recipients and environments, whether two parameters were specific to recipients (self vs other), whether two parameters were specific to environments (poor vs rich), or whether four parameters were specific to each condition (self/poor vs self/rich vs other/poor vs other/rich). We compared variations on the shape of the discounting function, including linear, parabolic, hyperbolic, and power functions[57,59]

$$\text{Linear}: \quad SV = (r*p) - o$$

$$\text{Parabolic}: \quad SV = (r*p)^2 - o$$

$$\text{Hyperbolic}: \quad SV = (r*p)/(1+o)$$

$$\text{Power}: \quad SV = (r*p)^\theta - o$$

In these models, the subjective value $SV$ of an opportunity results from comparing its reward magnitude $r$ and probability $p$ against the opportunity costs $o$ associated with accepting it. Higher values of $o$ will generally make people more selective and thus less likely to pursue an opportunity, because the expected value of the reward is further reduced by the higher costs. Given the choice patterns in the GLMMs, we expected higher $o$ values in the rich environment. Lower values of $o$ suggest that people generally will be less selective and are more willing to act. In the power function model, the free parameter $\theta$ individually scales the value of the expected reward (magnitude*probability). We also compared variations of the value and weighting functions from prospect theory[58,61] that separately adjusted the scaling of the reward's magnitude and probability,

$$\text{Prospect}: \quad SV = v(r)*w(p) - o$$

where the value $v$ and weighting $w$ functions are defined as,

$$v(r) = r^\alpha$$

$$w(p) = \frac{p^\gamma}{(p^\gamma + (1-p)^\gamma)^{-\gamma}}$$

Here, $\alpha$ represents risk aversion, and $\gamma$ represents the weighting placed on the reward's probability. Finally, we also compared models that accounted for the running background reward rate[1]. Participants may have tracked the average expected reward in each block,

$$\text{Reward tracking}: SV = (r*p) - \rho - o$$

where $\rho$ represents the cumulative average of the reward rate (magnitude * probability) on each trial within a block. In total, we tested 128 different models that varied in shape and in the number of opportunity costs and decision noise parameters. All models were fit to choices using a logistic function,

$$\text{Pr(choose to act)} = \frac{1}{1 + e^{-(\beta*SV)}}$$

and estimated using an iterative maximum a posteriori (MAP) approach[89,90] in MATLAB (version 2022a). The probability of choosing to act, Pr(choose to act), is based on $SV$ and a value sensitivity (inverse temperature) parameter $\beta$.

## Model identifiability and parameter recovery

We used simulated data to ensure the validity of our model comparison and parameter fitting procedures[91]. For model identifiability, we used each model to separately simulate choice data for 100 artificial agents. Simulated parameters were drawn from a uniform distribution with lower and upper bounds matched to the range of parameter values observed in the real participants. The simulated data were separately fit to each model using the same MAP procedure as was used on the real participants' data. This allowed us to create a confusion matrix of model exceedance probabilities, which showed good identifiability (Fig. 5a). Exceedance probabilities were calculated using the 'spm_BMS' function from SPM 12. For parameter recovery, we used the winning model to simulate data for 5000 artificial agents, again drawing the parameters from a uniform distribution bounded by the range observed in the real participants. We fit the simulated data using the same MAP procedure as before and correlated the recovered parameter values with the 'true' simulated values. This showed that our winning model had good parameter recovery ($rs > 0.80$; Fig. 5b).

## Exploratory factor analysis

We conducted an exploratory factor analysis on the subscales of the questionnaires using the 'psych' package (version 2.3.6) for R. We included all the subscales from the AMI and OUS, only the depression and anxiety subscales from the DASS, and the cognitive empathy and affective empathy subscales from the QCAE (Fig. 6a). Two participants did not complete all the questionnaires due to technical issues and therefore were not included in the analysis. To extract the factor loadings, we used maximum likelihood and an oblique rotation (oblimin). Parallel analysis and inspection of the scree plot suggested a 3-factor solution. These 3 factors captured 54.15% of the variance in the set of measures and were minimally correlated (the highest $r = 0.12$). Participant-level scores for each factor were computed using Thurstone's regression method. These scores were then correlated with the fitted parameters from the winning model using Pearson's $r$, and $p$-values were FDR adjusted using the Benjamini–Hochberg procedure.

## Reporting summary

Further information on research design is available in the Nature Portfolio Reporting Summary linked to this article.

## Data availability

The raw datasets analysed during the current study are available at: https://doi.org/10.17605/osf.io/dmfhq[92].

## Code availability

The code used for data analysis, visualisation, and model fitting are available at: https://doi.org/10.17605/osf.io/dmfhq[92].

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

## Acknowledgements

This work was supported by a Medical Research Council Fellowship (MR/P014097/1 and MR/P014097/2), a Christ Church Junior Research Fellowship, a Christ Church Research Centre Grant, a Jacobs Foundation Research Fellowship, a Leverhulme Prize (PLP-2021-196), a Wellcome Trust/Royal Society Sir Henry Dale Fellowship (223264/Z/21/Z), and a UKRI EPSRC Frontiers Research Guarantee/ERC Starting Grant (EP/X020215/1) to P. L. Lockwood. We would like to thank Marco Wittmann for helpful feedback and guidance on early findings from the studies described here.

## Author contributions

Conceptualisation: T.A.V., L.P., M.F.S.R. and P.L.L. Methodology: T.A.V., L.P., J.C., M.F.S.R. and P.L.L. Investigation: T.A.V., L.P. and T.H. Formal analysis: T.A.V., L.P., J.C. Writing—original draft: T.A.V. and P.L.L. Writing—review and editing: T.A.V., J.C., N.K., N.G., M.A.J.A., M.F.S.R. and P.L.L. Funding acquisition: P.L.L. Supervision: M.R. and P.L.L.

## Competing interests

The authors declare no competing interests.
