## [Transparent Peer Review file · Nature Communications]

Humans are more prosocial in poor foraging environments

Corresponding Author: Dr Todd Vogel

Version 0:

Reviewer comments:

Reviewer #2

(Remarks to the Author)

Thanks to the authors for taking my comments into account. Overall, I think you've done a solid job responding to both my points and those raised by other reviewers.

Broadly, my concerns fall into two areas.

First, I was initially unsure whether the key interaction effect between environment and recipient might simply be due to the way the rewards were set up—especially if the rewards across environments weren't fully balanced. The analysis focusing on the central cells helped clarify this, and I found that part convincing. I'm persuaded by your defense of the effect there. The second part of my concern is more about the use of the term opportunity cost in the paper, which I think still needs some clarification. Here's why:

1. How opportunity cost is defined: Around line 772, you describe ρ as capturing how the current option compares to alternatives people could pursue. But this feels a bit different from how opportunity cost is usually understood. Typically, it's about how much you lose out on when you skip a good opportunity. That would mean in a rich environment, where good options are everywhere, missing one doesn't cost much—so low opportunity cost. In contrast, in a poor environment, good options are rare, so missing one hurts more—higher opportunity cost. That's kind of the opposite of how ρ is being used here. I understand models can define constructs in different ways, but it'd be helpful to back this up with some references or theoretical grounding.

2. What exactly opportunity cost is comparing to: In your response, you said ρ compares the current option to other options—not to the movie. But in your extra analysis, you mention the movie is enjoyable, which could also lower the value of choosing something else (like pressing the spacebar). If that's the case, shouldn't the movie's value be in the model too? Right now, it feels like there's a bit of a contradiction here. The experiment clearly sets up a "to-do vs. doing" trade-off (watching the movie vs. taking action), but the model doesn't really reflect that, which makes it harder to see the novelty in how the model ties into the task design.

3. Learning effects: I do think the environment has an impact—people likely evaluate the same option differently in rich vs. poor environments. I noticed in Figure S3 that your full prospect model (e.g., M6 and M16) performs about the same as the risk aversion models, which suggests people might have different value of reward and probability. Participants had tendency to learning the different value during different environment trial-by-trial. But the model doesn't account for any learning process. If you want to argue against this idea, it might be helpful to include some analysis looking at how people's choices change over time.

4. Oversimplification of decision factors: It's also possible that participants are taking a lot of things into account: how good the current offer is (SV), how fun the movie is, how tiring the effort task is (which could change with time), and how good future offers might be. But right now, all of that seems to be wrapped into a single parameter ρ , which may be oversimplifying things a bit. It would be great if the model could tease these apart more clearly.

Minor issue:

1. It might be worth bringing in some ideas from scarcity theory to help interpret the findings.

2. Also, unless there's a strong theoretical reason for comparing $value_po$ and $value_rs$, that contrast might come off as a bit arbitrary. Readers might wonder if it's just an environment effect causing the no difference or recipient effect causing the no difference, which makes it harder to interpret the results.

Reviewer #3

(Remarks to the Author)

The authors have fully addressed all my previous comments, and are to be commended on a detailed and thorough revision of this manuscript.

Version 1:

Reviewer comments:

Reviewer #2

(Remarks to the Author)

The authors have fully addressed all my concerns. Congrats on this wonderful work!

REVIEWER COMMENTS

Reviewer #2 (Remarks to the Author):

Thanks to the authors for taking my comments into account. Overall, I think you've done a solid job responding to both my points and those raised by other reviewers. Broadly, my concerns fall into two areas.

Response: Thank you for taking the time to review our changes. We are delighted to hear that you found our revisions solid, and we have now further edited the manuscript in response to your additional feedback. In particular, we have included additional theoretical grounding and explanation, as well as a new set of computational models showing that learning effects did not account for participant behaviour.

First, I was initially unsure whether the key interaction effect between environment and recipient might simply be due to the way the rewards were set up—especially if the rewards across environments weren't fully balanced. The analysis focusing on the central cells helped clarify this, and I found that part convincing. I'm persuaded by your defense of the effect there.

Response: Thank you for your positive feedback on our additional analysis.

The second part of my concern is more about the use of the term opportunity cost in the paper, which I think still needs some clarification. Here's why:

R2.1 How opportunity cost is defined: Around line 772, you describe o as capturing how the current option compares to alternatives people could pursue. But this feels a bit different from how opportunity cost is usually understood. Typically, it's about how much you lose out on when you skip a good opportunity. That would mean in a rich environment, where good options are everywhere, missing one doesn't cost much—so low opportunity cost. In contrast, in a poor environment, good options are rare, so missing one hurts more—higher opportunity cost. That's kind of the opposite of how o is being used here. I understand models can define constructs in different ways, but it'd be helpful to back this up with some references or theoretical grounding.

Response: Thank you for the opportunity to clarify our definition of opportunity costs further. We agree that the term opportunity cost and model parameterisations can be used in slightly different ways. We defined opportunity costs based on work in behavioural ecology (Charnov, 1976; Krebs et al., 1974; Stephens & Krebs, 1986) describing how animals will adjust their foraging behaviour based on the quality of the environment, as well as more recent work in humans (e.g., Constantino & Daw, 2015; Khalighinejad et al., 2021; Garrett & Daw, 2020). In particular, we used the theoretical framework of prey selection within optimal foraging theory (Charnov, 1976; Krebs et al., 1977). Here, whether an organism selects its prey is based on the quality of the

prey encountered and the average quality of other prey in the environment. If most prey are good (e.g., in our task's rich environment where most opportunities are high in reward and high in probability) the opportunity cost of selecting the encountered prey is high because other, potentially better, opportunities are abundant in the environment (selecting the prey takes time and effort, which may be better used for other opportunities). In contrast, if most prey are bad (e.g., in our task's poor environment where most opportunities are low in reward and probability), then the opportunity cost of selecting the encountered prey is lower because future opportunities are less likely to be good. This leads to a higher acceptance rate of opportunities in the poor environment; the costs of pursuing a given opportunity are less when the alternatives are not expected to be better.

Using this conceptualisation, we set up our models to compute the subjective value SV of an opportunity by comparing its reward magnitude r and probability p against the opportunity costs o associated with accepting it. Higher values of o will generally make people more selective and thus less likely to pursue an opportunity, because the expected value of the reward (magnitude*probability) is reduced more by the higher costs. This is reflected in the overall higher o parameter in rich environments that we observed in our data. Lower values of o suggest that people generally will be less selective and therefore more willing to act. Our findings showed an overall lower o parameter in poor environments. We realise this definition is indeed opposite to the one that you outline; we hope to have now clarified our conceptualisation of opportunity cost in this prey-selection framework, supported by additional references and text changes.

Specifically, we have edited the Introduction to make the precise theoretical grounding clearer:

“Theories of ‘prey-selection’ in animal foraging suggest that the quality, or richness, of an animal’s environment determines how it chooses its prey (Charnov, 1976; Krebs et al., 1977). In support, a seminal study in birds showed that in rich environments, where the average quality of prey is higher, animals will reject low-quality prey they encounter and instead selectively wait for higher quality options (Krebs et al., 1977). Therefore, the opportunity costs of selecting prey are higher, as alternative options are likely to be of high quality too. When the environment is poor—i.e., where the average quality of prey is lower—waiting for high-quality options can be disadvantageous. The animals will begin to accept low-quality prey because future opportunities are not expected to be better (i.e., the opportunity cost of selecting a prey is less, because better alternatives are not readily available).”

We have also updated the text in the Computational Modelling section of the Methods:

“In these models, the subjective value SV of an opportunity results from comparing its reward magnitude r and probability p against the opportunity costs (o) associated with

accepting it. Higher values of o will generally make people more selective and thus less likely to pursue an opportunity, because the expected value of the reward is further reduced by the higher costs. Given the choice patterns in the GLMMs, we expected higher o values in the rich environment. Lower values of o suggest that people generally will be less selective and therefore more willing to act.”

R2.2 What exactly opportunity cost is comparing to: In your response, you said o compares the current option to other options—not to the movie. But in your extra analysis, you mention the movie is enjoyable, which could also lower the value of choosing something else (like pressing the spacebar). If that's the case, shouldn't the movie's value be in the model too? Right now, it feels like there's a bit of a contradiction here. The experiment clearly sets up a "to-do vs. doing" trade-off (watching the movie vs. taking action), but the model doesn't really reflect that, which makes it harder to see the novelty in how the model ties into the task design.

Response: Thank you for the chance to clarify further. Our computational models were inspired by previously established models of prey selection from behavioural ecology (Krebs et al., 1977). These models state that the decision to accept or reject prey opportunities depends on the quality of the prey (i.e., its reward value) as well as the average quality of prey in the wider environmental context. To examine these decisions, we built upon discounting models, since they are commonly used to study value-based decision-making in behavioural economics and decision neuroscience (e.g., Hartmann et al., 2013; Klein-Flügge et al., 2015; Madden & Bickel, 2010; Myerson & Green, 1995; Ostaszewski et al., 2013). These models compute how subjective value (e.g., how the value of option A compares to option B) is used to determine how people choose between different options. Our model instead uses subjective value to determine how people choose to accept or reject a single option they encounter.

In our model, participants are not necessarily choosing between the movie and the opportunity, but rather deciding *when* they want to interrupt their ongoing behaviour to start another action. To make the task more ecological we chose to play a movie in the background to reflect the kinds of 'when to act' decisions made in everyday life. We felt this was important compared to, for example, using a grey background screen, which is common in most psychological experiments. This helped ensure that participants were not selecting opportunities simply because they were bored or because of the demand characteristic to just do something. Subjective value is computed from the value of the reward discounted by the costs associated with pursuing that opportunity, rather than a direct comparison between the value of the reward and the value of the movie. Additionally, the value of the movie is consistent across blocks; this allowed the separate o parameters to capture differences driven by changes in the environment and recipient specifically. We have updated the Discussion to include additional rationale for the role of the video in our design accordingly:

“In addition, by including a video as the baseline option rather than a plain screen, which is common in psychology experiments, we could measure behaviours in a more ecological context and try to rule out effects of boredom or demand characteristics of doing something on decisions to act.”

R2.3 Learning effects: I do think the environment has an impact—people likely evaluate the same option differently in rich vs. poor environments. I noticed in Figure S3 that your full prospect model (e.g., M6 and M16) performs about the same as the risk aversion models, which suggests people might have different value of reward and probability. Participants had tendency to learning the different value during different environment trial-by-trial. But the model doesn't account for any learning process. If you want to argue against this idea, it might be helpful to include some analysis looking at how people's choices change over time.

Response: Thank you for the suggestion, we agree that learning may play an interesting and important role in our findings. In our generalised linear mixed-effects models we already accounted for potential learning effects by examining trial number and previous expected values. Across all studies, we found that participants were less likely to accept an opportunity in later trials. In studies 2 and 3 we found that participants were less likely to accept an opportunity if the expected value on the previous opportunity was higher (the effect in study 1 was in the same direction but was not statistically significant ($p = 0.097$)). After accounting for these effects, we still observed a significant 3-way interaction between environment, recipient, and expected value, providing further evidence that our main findings were not driven by learning.

The above analyses, however, may not adequately capture learning effects across trials and within blocks. For example, participants may track the ongoing background reward rate and use that as a reference to compare the current opportunity against (Constantino & Daw, 2015; Gabay & Apps, 2021). To examine this, we have now tested a new set of computational models that included a parameter that tracked the background reward rate from trial-to-trial. Specifically, the models included an additional parameter of the cumulative average reward rate within each block. Like all other models, these reward tracking models also included parameters for opportunity cost (α) and value sensitivity (β) that differed based on environment and recipient (16 unique models).

Comparing this new set of models, we found that they had significantly worse fits compared to our original winning model (see Figure S3 in supplementary materials). Examining BICint scores shows that these new models perform poorly compared to the other types of models. Our original winning model still showed the highest exceedance probability amongst all the models (95.8%).

We have now added this additional set of models to the Methods and Supplementary Materials:

Added to the Computational Modelling section of the Methods:

“Finally, we also compared models that accounted for the running background reward rate¹. Participants may have tracked the average expected reward in each block,

$$\text{Reward tracking: } SV = (r * p) - \rho - o$$

where ρ represents the cumulative average of the reward rate (magnitude * probability) on each trial within a block.”

We have also updated Figures S2 and S3 in the supplementary material:

Figure S2:

Figure S3:

R2.4 Oversimplification of decision factors: It’s also possible that participants are taking a lot of things into account: how good the current offer is (SV), how fun the movie is, how tiring the effort task is (which could change with time), and how good

future offers might be. But right now, all of that seems to be wrapped into a single parameter α , which may be oversimplifying things a bit. It would be great if the model could tease these apart more clearly.

Response: Thank you for your comment. We agree that there are likely several factors that participants are taking into account. However, our findings suggest that the critical factors that influence decisions are the environment, the recipient, and the expected value of the reward. Our model comparisons showed that separate opportunity cost parameters for environment and recipient best explained participants' choice behaviour. This suggests that these two factors influenced participants' choices more so than other factors, such as the salience of the movie or the experience of the effort task (which did not vary systematically across environments or recipients). However, we acknowledge that these other factors may exert greater influence in other contexts (e.g., varying effort levels).

If environment and recipient played a reduced or no role in decisions, we would expect to find a single α parameter to best explain behaviour. In addition, the advantage of our computational modelling approach was to use the softmax to calculate choice consistency/sensitivity to value (cf. Krebs et al., 1977). The analysis of the softmax provides additional insight into factors influencing choice, as it captured how people were equally value-driven in other-poor and self-rich environments. This is particularly interesting as it is a frequent finding that people are consistently less value sensitive when deciding for others in previous research (Lockwood et al., 2017; 2022; 2024).

We also rule out the influence of other decision factors in our mixed-effects models—trial number, previous expected value, and previous choice. After controlling for these variables, and paralleling the results of the computational model, the main influences on choice were the environment, recipient, and expected value.

For these reasons, and also due to the robustness of our model fit against alternatives with simplified decision processes, we are confident that our model accurately captures the influences on choice behaviour. Future studies would be needed that manipulate how tiring the effort task is, or how enjoyable the movie is, to quantify the specific influence of these variables.

Based on our response above, we have added to the Discussion:

“This finding is particularly interesting as the majority of studies testing how we decide for others in the context of effort and rewards suggest that we are less value sensitive for others compared to ourselves (Lockwood et al., 2017; 2022; 2024).”

“In addition, by including a video as the baseline option rather than a plain screen, which is common in psychology experiments, we could measure behaviours in a more

ecological context and try to rule out effects of boredom or demand characteristics of doing something on decisions to act.”

Minor issue:

R2.5 It might be worth bringing in some ideas from scarcity theory to help interpret the findings.

Response: Thank you for this suggestion. We have elaborated on the potential role of scarcity theory in the Discussion within the confines of word count considerations:

“How these environmental factors translate to one’s broader socioeconomic environment remains to be investigated. Different social psychological theories could inform and be informed by our findings. For example, scarcity theory proposes that being in a poor environment induces a scarcity mindset (de Bruijn & Antonides, 2022), which results in suboptimal decisions and behaviours. Here we suggest that poor environments encourage people to act to help others, whether this behaviour can be viewed as optimal is a question for future studies.”

R2.6 Also, unless there’s a strong theoretical reason for comparing value_po and value_rs, that contrast might come off as a bit arbitrary. Readers might wonder if it’s just an environment effect causing the no difference or recipient effect causing the no difference, which makes it harder to interpret the results.

Response: Thank you for this suggestion. Also in response to your point R2.4, we believe this finding is robust (there is strong Bayesian evidence of no difference between these two conditions) and would be of interest to readers in the field. Most studies on decision-making that compare deciding for self and other find lower value sensitivity when making decisions for someone else compared to self in learning and in effort-based decision-making (Lockwood et al., 2017; 2022; 2024). We therefore believe it is notable to highlight a context in which value sensitivity is manipulated to be equal for self and other, particularly because the more surprising behaviour we observe is when individuals are in poor environments and deciding for others. We have made this clearer in the Discussion:

“This finding is particularly interesting as the majority of studies testing how we decide for others in the context of effort and rewards suggest that we are less value sensitive for others compared to ourselves (Lockwood et al., 2017; 2022; 2024).”

Reviewer #3 (Remarks to the Author):

The authors have fully addressed all my previous comments, and are to be commended on a detailed and thorough revision of this manuscript.

Response: Thank you for your time re-reviewing our manuscript and your very useful feedback that has helped us to improve it. We are delighted that you found our revision thorough and detailed.